# Optimal Nutrient Solution and Dose for the Yield of Nuclear Seed Potatoes under Aeroponics



**Jaime B. Silva Filho** [1,*] , **Paulo Cezar Rezende Fontes** [2,†], **Jorge F. S. Ferreira** [3,*], **Paulo R. Cecon** [4] and **Elizabeth Crutchfield** [5]

1 Department of Microbiology and Plant Pathology, University of California Riverside, 900 University Avenue, Riverside, CA 92521, USA
2 Department of Agronomy, Federal University of Viçosa, Av. Peter Henry Rolfs, s/n, UFV Campus, Viçosa 36570-000, MG, Brazil
3 US Salinity Laboratory, USDA-ARS, 450 W. Big Springs Rd., Riverside, CA 92507, USA
4 Department of Statistics, Federal University of Viçosa, Av. Peter Henry Rolfs, s/n, UFV Campus, Viçosa 36570-000, MG, Brazil
5 Department of Botany and Plant Sciences, University of California Riverside, 900 University Avenue, Riverside, CA 92521, USA
* Correspondence: jaimeufv@gmail.com (J.B.S.F.); jorge.ferreira@usda.gov (J.F.S.F.)
† CNPq Fellow, Brazil (Proc. 303448/2018-0).

**Abstract:** The aeroponic production of certified seed potatoes is a booming alternative for arid and semi-arid areas where fresh water is scarce and soil-borne diseases and nematodes preclude field production. Although widely used in aeroponics, nutrient-solution salinity effects have not been evaluated in potatoes. This study aimed to (1) establish the best of two nutrient solutions (Otazú vs. modified Furlani) at 20, 50, 100, and 150% of the crop-recommended dose for seed-potato production, (2) evaluate growth indexes to diagnose plant-N status, and (3) establish a prognosis for the yield of nuclear seed potatoes under aeroponics. At 21 days after transplanting, there was a significant correlation between the nitrate-N petiole-sap test and some of the parameters measured. The 4th leaf indexes correlated with yield parameters indicating that they can be used to prognosticate the final minituber yield. The best parameters to diagnose the N status in potato plants were: 4th leaf area, length, and dry weight (Otazú's), SPAD, and 4th leaf area (modified Furlani's). Although both nutrient solutions had similar nitrogen concentrations, Otazú's nutrient solution at 100% of the recommended nitrogen dose had lower salinity than the modified Furlani's solution and was the best to produce nuclear seed potatoes.

**Keywords:** fertilization; 4th leaf; SPAD; NBI; nitrogen; salinity; electrical conductivity

## 1. Introduction

One of the challenges of the aeroponics technique is that the solution must provide the roots with water and all the essential nutrients for plant growth, development, and tuber production, including seed-potato minitubers. Additionally, the nutrient solution must be balanced to avoid competition among macronutrients (e.g., K vs. Ca) and between essential mineral nutrients with excessive salt ions of Na and Cl (e.g., $Na^+$ vs. $K^+$ and $Cl^-$ vs. $NO_3^-$). As a major macronutrient source, nutrient solutions must be adjusted accordingly to prevent N deficiency and achieve maximum productivity of seed potatoes [1,2]. The nutrient solution needs to be adapted to the cultivar and the plant growth phases, specifically the composition of the N solution which, in hydroponics, can vary from 3.0 to 19.0 mmol $L^{-1}$ [3–6]. For the aeroponic system, the N concentration from different sources was reported to range from 13.5 to 14.2 mmol $L^{-1}$ of N [7,8]. A five-year study on irrigated potato production systems in a sandy loam soil evaluated nitrogen (N) sources such as ammonium sulfate, ammonium nitrate, urea, and calcium nitrate to maximize potato yield [9].

In the end, Bundy and co-workers [9] concluded that ammonium nitrate was the best N source for potato production in an irrigated system. Silva et al. [10] evaluated calcium nitrate and urea as N sources in a hydroponic system. Contrary to Bundy et al. [9], they concluded that the nitric, rather than the ammoniacal, source was more suitable for potato production in hydroponic systems.

In any well-managed production process, the expected occurrence of visual symptoms of nitrogen deficiency or toxicity is small, but inadequate concentrations of N may hinder the effective production of tubers without any visual symptoms of N deficiency. Under this scenario, nitrogen deficiency or toxicity is difficult to assess quantitatively only through appearance, being more reliably quantified by conducting mineral analyses [11,12].

There are several indexes, or variables, for the monitoring of the nutritional status of N in plants, the most traditional being the chemical analysis to establish the N concentration (g/100 g or %) in dry leaves [12–14]. This test requires a well-equipped laboratory, qualified personnel for sample preparation, time, and expertise to run the costly analytical equipment. This traditional method can be replaced by faster tests, such as the ionic composition of the aeroponics water, with the advantages of correcting eventual deficiencies of N during the crop cycle and acquiring the nutritional status of the plant in real time [15,16].

Indexes to evaluate the N status in a plant can be obtained in real-time and include the biometric variables of the plant [17]. For example, the area of the fourth leaf, total leaf area, dry matter of root, stem, and leaf samples, leaflet numbers of the fourth leaf, total number of leaves of the plant, length and diameter of the stem, and the dry matter mass of the fourth leaf [18–20]. Additionally, there is the possibility of quantifying the $NO_3^-$-N level in the petiole sap and the intensity of the green or chlorophyll with equipment such as a DUALEX Scientific, SPAD, and a leaf color chart [11,12,21–23].

As for all index-based diagnostics, it is necessary to have a critical value considered appropriate for each index [24], thus becoming necessary to calibrate and adjust the critical value of each index, in the present study, aeroponics, and multiplication by sprouts.

To date, there are no reports establishing the nitrogen concentrations in plants of nuclear seed potatoes, cloned through sprouts, and under aeroponics. Additionally, the effect of acute and chronic deficiency of N on nuclear seed-potato production under aeroponics has not been evaluated.

Thus, this study used nuclear seed-potato plants, generated by cloning, under aeroponics, to (1) establish the best of two nutrient solutions (Otazú vs. modified Furlani) at 20, 50, 100, and 150% of the crop-recommended dose for seed-potato production, (2) evaluate growth indexes to diagnose plant-N status, and (3) establish a prognosis for the yield of nuclear seed potatoes under aeroponics.

## 2. Materials and Methods

### 2.1. Location and Conditions

A greenhouse experiment was conducted from April to July 2014, during two seasons: fall (March–May) and winter (June–August), at the Federal University of Viçosa, Minas Gerais, Brazil (20°45′27.05″ S 42°52′11.63″ W, 649 m above sea level). Figure 1 depicts the daily, maximum, and minimum temperatures and relative humidity during the experiment.

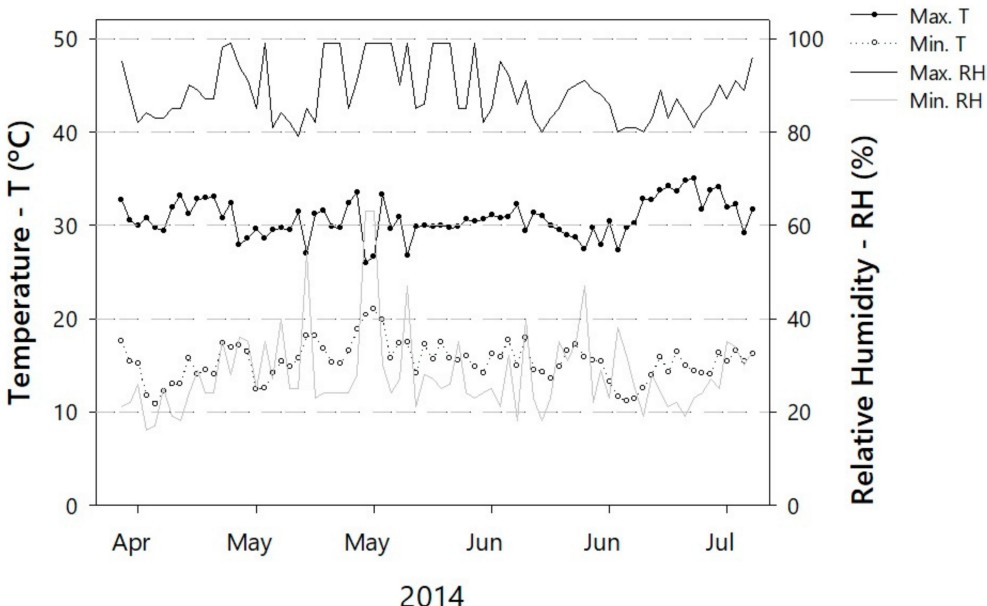

**Figure 1.** Daily maximum and minimum temperatures and relative humidities as measured inside the greenhouse where nuclear seed potatoes were grown under aeroponics from April to July 2014.

### 2.2. Design

The study was set up in a randomized complete block design (RCBD) with nine replications. This consisted of two nutrient solutions: Otazú [8] vs. one modified from Furlani [3] at 20%, 50%, 100%, and 150% of the crop-recommended dose to produce seed potatoes, in a factorial arrangement $4 \times 2$, $n = 72$.

### 2.3. Aeroponic System

The aeroponic system was proposed by Otazú [8] and sprouts were used as cloned propagation material. The sprouts (8 cm long) were collected from the seed tubers of the pre-nuclear category (virus-free) cv. Agata (Figure 2D,E). Sprouts were then immersed in 2% sodium hypochlorite solution for 2 min, then washed with tap water for 5 min [25]. Then, they were planted in black Deepot$^{TM}$ cells with dimensions of 50 mm $\times$ 178 mm and a volume of 262 mL (D16H, https://www.stuewe.com/products/deepots.php, accessed on 26 October 2022) containing Tropstrato HT Hortaliças$^{®}$ until rooted. Tropstrato HT Hortaliças$^{®}$ is made up of pine bark, vermiculite, peat, N-P-K fertilizer (14-16-18), potassium nitrate (13% N and 44% $K_2O$, salt index 69.5), and simple superphosphate (https://www.casadasementeslavras.com.br/tropstrato-ht-hortalicas-25kg-vida-verde, accessed on 26 October 2022). After 16 days of planting, uniform seedlings, with roots and an average length of 18 cm, were washed for removal of the substrate. The wood boxes were made with the following dimensions: 1.0 mL $\times$ 0.6 mW $\times$ 0.7 mH (Figure 2A). For all treatments, we used the MA-30 nozzle, which has a flow rate of 34 L h$^{-1}$ and produces droplets of 100 μm at 3 atm. Nutrient solutions were applied using two downward misting nozzles (MA-30$^{®}$ Agrojet, red nozzle color without anti-drip (https://www.agrojet.com.br/produto/ma-30-com-base-de-rosca-1-4/, accessed on 26 October 2022) spaced 0.3 m apart in the box. The walls of the box were coated with 50 mm Styrofoam sheets with access windows that allowed for the harvesting of tubers. The interior of the box was covered with black plastic to exclude light. The irrigation system consisted of a 60-L bucket, a $\frac{1}{2}$-hp electric water pump, PVC pipes (25 mm diameter), and misting nozzles. Throughout the experiment, the pumps were controlled by a digital timer that alternated between 20 s of misting and 3 min idle. We determined the time for the nutrient solution spray intervals based on a previous study. There are different published misting protocols, such as 15 min on/15 min off [8,26,27], 10 s every 20 min [28], 30 s every 5 min [29], 20 s every 5 min [30], and 10 s every 10 min [31]. The length of time the nutrient sprays the solution on plant

roots is determined by several factors, including the type of pump used and the volume of the nutrient-solution reservoir. When we tested spray intervals of 15 min on/15 min off, we discovered that the nutrient solution overheated, causing damage to the potato plant. Finally, we determined that a spray interval of 20 s every 3 min was the most adequate for producing nuclear seed potatoes. The best nozzle type and spray direction were previously established [32] because data for these parameters were not available at the time. We chose the MA-30® nozzle due to its availability and low cost. Holes were created on the cover of boxes to accommodate the installation and growth of transplanted seedlings (Figure 2B). The seedlings were transplanted into holes on the Styrofoam lids of the wood boxes (Figure 2C,F). The developmental and production stages of seed-potato minitubers in a CIP aeroponic system are depicted in Figure 2E–H.

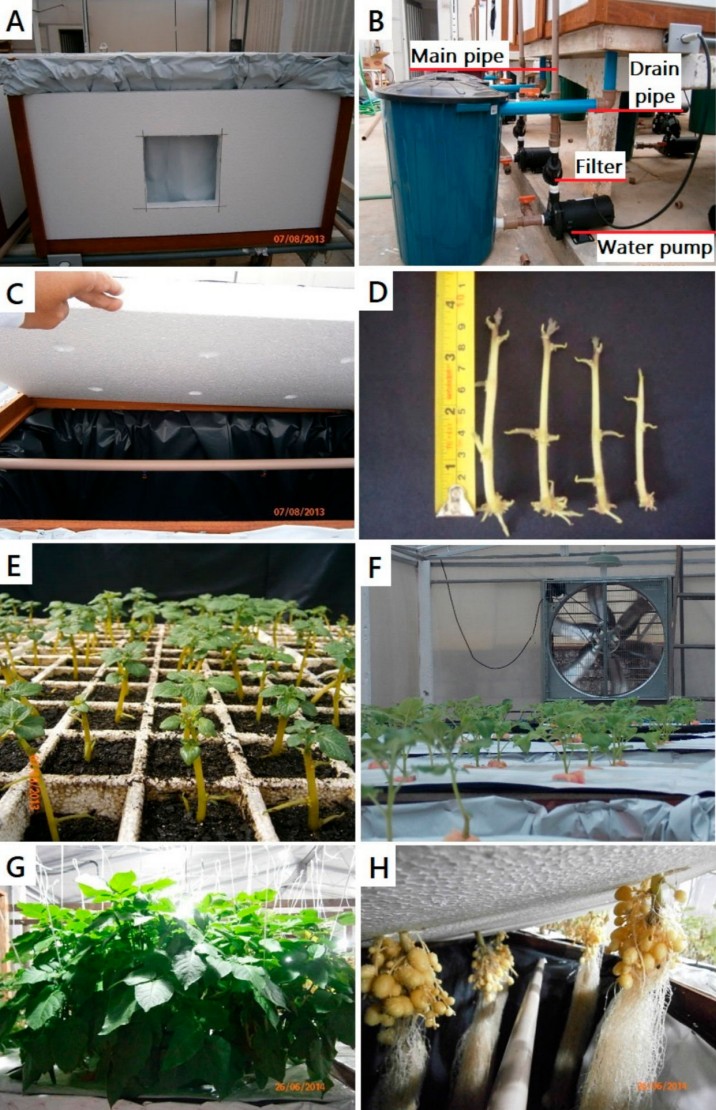

**Figure 2.** Wood box (**A**); connection to a nutrient solution bucket, electric water pump, filter, and main pipe that irrigate plants in the box, and a drain pipe that recycles the nutrient solution (**B**); holes in the upper Styrofoam box lid (**C**); (**D**) virus-free sprouts from pre-nuclear seed-potato plants that were positioned through the holes pictured in (**C**) to originate plants as a source of potato seeds; Rooting of sprouts (**E**); growth of potato plants (**F**); potato plant tutoring with twine string (**G**); seed-potato minitubers (**H**).

## 2.4. Nutrient Solutions

The nutrient solutions used were either a modification of Furlani's [3] or proposed by Otazú [8] (Table 1), and consisted of the following salts and their salt indexes (SI) calculated (only for macronutrients) based on the osmotic pressure generated by the same mass of $NaNO_3$ (SI = 100): $Ca(H_2PO_4)_2$ (neglectable SI), $MgSO_4$ $7H_2O$ (SI = 44), $NH_4NO_3$ (SI = 104), $KNO_3$ (SI = 74), KCl (SI = 116.2), $NaNO_3$ (SI = 100), $ZnSO_4$ $7H_2O$, $MnSO_4$ $H_2O$, $H_3BO_3$, $(NH_4)_6Mo_7O_{24}$ $4H_2O$, $FeCl_3$ $6H_2O$, and $C_{10}H_{14}N_2O_8Na_2$ $2H_2O$ (EDTA) for Otazú (2010); and $KH_2PO_4$ (SI = 8.4), $MgSO_4$ $7H_2O$ (SI = 44), $NH_4NO_3$ (SI = 104), $Ca(NO_3)_2$ $4H_2O$ (SI = 53), $KNO_3$, $(NH_4)_2SO_4$ (SI = 68.3), $CaCl_2$ $2H_2O$ (SI = 82), KCl (SI = 116.2), $NaNO_3$ (SI = 100), $K_2SO_4$ (SI = 42.6), $CuSO_4$, $ZnSO_4$ $7H_2O$, $MnSO_4$ $H_2O$, $H_3BO_3$, $(NH_4)_6Mo_7O_{24}$ $4H_2O$, $FeCl_3$ $6H_2O$, and $C_{10}H_{14}N_2O_8Na_2$ $2H_2O$ (EDTA) for the modified Furlani [3]. Following Otazú [8] recommendations, the ammonium nitrate concentration was reduced by half after 22 DAT (Table 1). This reduction is shown in Table 2 for ammonium only. A cation-anion balance was performed for the nutrient solutions used in this study. The individual concentrations of mineral nutrients used, at each of the four concentrations tested, and the electrical conductivity of each nutrient solution at each concentration is given in Table 1. Electrical conductivity measurements, pH monitoring, and nutrient solution replacement were done according to Silva Filho et al. [33]. For the nutrient solution proposed by Otazú [8], the electrical conductivities (EC) were 1.10, 1.26, 1.60, and 2.12 dS $m^{-1}$ up to 21 days after transplanting (DAT) for the N concentrations of 20%, 50%, 100%, and 150% of the standard (crop-recommended) N concentration, respectively. From 22 to 61 DAT, EC were 1.00, 1.15, 1.35, and 1.80 dS $m^{-1}$ for N concentrations of 20%, 50%, 100%, and 150% of the standard N concentration, respectively (Table 1). For the nutrient solution modified from Furlani [3], Up to 21 DAT, the EC were 1.93, 2.00, 2.11, and 2.72 dS $m^{-1}$ for N concentrations of 20%, 50%, 100%, and 150% of the standard N concentration, respectively. From 22 to 61 DAT, EC were 1.85, 2.00, 2.30, and 2.65 dS $m^{-1}$ for N concentrations of 20%, 50%, 100%, and 150% of the standard N concentration, respectively (Table 1). The salt mixes used to achieve the different concentrations (20%, 50%, 100%, and 150%) of the standard N concentration proposed by either Furlani or Otazú are given in Table 2. We used deionized water to prepare all the nutrient solutions. If tap water would be used, consider that the salinity of each solution would increase by an average of 0.7 dS $m^{-1}$ due to the chlorination of potable water. The potato crop is classified as moderately sensitive to salinity with a threshold for the saturated soil extract (ECe) of 1.7 dS $m^{-1}$ and irrigation water salinity (ECw) of 1.1 dS $m^{-1}$ [34].

**Table 1.** Salts reported in nutrient solutions for stock solutions were used to produce nuclear potato seeds in an aeroponic system.

| Treatments | SALTS |
|---|---|
| | Otazú's Nutrient Solution |
| 1 (20% of standard N concentration) | $MgSO_4$ $7H_2O$, $NH_4NO_3$, $Ca(H_2PO_4)_2$ $H_2O$, $KNO_3$, and KCl |
| 2 (50% of standard N concentration) | $MgSO_4$ $7H_2O$, $NH_4NO_3$, $Ca(H_2PO_4)_2$ $H_2O$, $KNO_3$, and KCl |
| 3 (100% of standard N concentration) | $MgSO_4$ $7H_2O$ (SI = 44), $NH_4NO_3$ (SI = 104), $Ca(H_2PO_4)_2$ $H_2O$, and $KNO_3$ (SI = 74) |
| 4 (150% of standard N concentration) | $MgSO_4$ $7H_2O$, $NH_4NO_3$ (SI = 104), $Ca(H_2PO_4)_2$ $H_2O$, $KNO_3$ (SI = 74), and $NaNO_3$ (SI = 100) |
| | Modified Furlani's Nutrient Solution |
| 1 (20% of standard N concentration) | $KH_2PO_4$, $MgSO_4$ $7H_2O$, $NH_4NO_3$, $(NH_4)_2SO_4$, $K_2SO_4$, $Ca(NO_3)_2$ $4H_2O$, $CaCl_2$ $2H_2O$, and KCl |
| 2 (50% of standard N concentration) | $KH_2PO_4$, $MgSO_4$ $7H_2O$, $NH_4NO_3$, $(NH_4)_2SO_4$ (SI = 68.3), $K_2SO_4$, $Ca(NO_3)_2$ $4H_2O$ (SI = 53), $CaCl_2$ $2H_2O$ (SI = 82), and $KNO_3$ (SI = 74) |
| 3 (100% of standard N concentration) | $KH_2PO_4$ (SI = 8.4), $MgSO_4$ $7H_2O$ (SI = 44), $NH_4NO_3$ (SI = 104), $(NH_4)_2SO_4$ (SI = 68.3), $Ca(NO_3)_2$ $4H_2O$ (SI = 53), $NaNO_3$ (SI = 100), KCl (116.2), and $KNO_3$ (SI = 74) |
| 4 (150% of standard N concentration) | $KH_2PO_4$ (SI = 8.4), $MgSO_4$ $7H_2O$ (SI = 44), $NH_4NO_3$, $(NH_4)_2SO_4$, $Ca(NO_3)_2$ $4H_2O$, $NaNO_3$, and $KNO_3$ |

Salt micronutrient solutions: $FeCl_3$ $6H_2O$, $C_{10}H_{14}N_2O_8Na_2$ $2H_2O$ (EDTA), $MnSO_4$ $H_2O$, $H_3BO_3$, $ZnSO_4$ $7H_2O$, $CuSO_4$, and $(NH_4)_6Mo_7O_{24}$ $4H_2O$.

**Table 2.** Nutrient concentrations in nutrient solutions tested in an aeroponic system to produce nuclear potato seeds.

| Nutrient | Otazú's Nutrient Solution | | | | | | | |
| --- | --- | --- | --- | --- | --- | --- | --- | --- |
| | Up to 21 DAT | | | | 22 to 61 DAT | | | |
| | 20% | 50% | 100% | 150% | 20% | 50% | 100% | 150% |
| | $mmol_c \, L^{-1}$ | | | | | | | |
| Nitrate | 1.96 | 4.9 | 9.80 | 14.70 | 1.52 | 3.80 | 7.60 | 11.40 |
| Ammonium | 0.88 | 2.2 | 4.40 | 6.60 | 0.44 | 1.10 | 2.20 | 3.30 |
| Phosphorus | 2.60 | 2.6 | 2.60 | 2.60 | 2.60 | 2.60 | 2.60 | 2.60 |
| Potassium | 5.40 | 5.40 | 5.40 | 5.40 | 5.40 | 5.40 | 5.40 | 5.40 |
| Calcium | 1.30 | 1.30 | 1.30 | 1.30 | 1.30 | 1.30 | 1.30 | 1.30 |
| Magnesium | 1.00 | 1.00 | 1.00 | 1.00 | 1.00 | 1.00 | 1.00 | 1.00 |
| Sulfur | 1.00 | 1.00 | 1.00 | 1.00 | 1.00 | 1.00 | 1.00 | 1.00 |
| | $\mu mol \, L^{-1}$ | | | | | | | |
| Iron (Chelated) | 151.97 | 151.97 | 151.97 | 151.97 | 151.97 | 151.97 | 151.97 | 151.97 |
| Manganese | 8.74 | 8.74 | 8.74 | 8.74 | 8.74 | 8.74 | 8.74 | 8.74 |
| Boron | 98.15 | 98.15 | 98.15 | 98.15 | 98.15 | 98.15 | 98.15 | 98.15 |
| Zinc | 2.75 | 2.75 | 2.75 | 2.75 | 2.75 | 2.75 | 2.75 | 2.75 |
| Copper | 2.83 | 2.83 | 2.83 | 2.83 | 2.83 | 2.83 | 2.83 | 2.83 |
| Molybdenum | 0.13 | 0.13 | 0.13 | 0.13 | 0.13 | 0.13 | 0.13 | 0.13 |
| EC ($dS \, m^{-1}$) | 1.10 | 1.26 | 1.60 | 2.12 | 1.00 | 1.15 | 1.35 | 1.80 |
| | Modified Furlani's Nutrient Solution | | | | | | | |
| | Up to 21 DAT | | | | 22 to 61 DAT | | | |
| | 20% | 50% | 100% | 150% | 20% | 50% | 100% | 150% |
| | $mmol_c \, L^{-1}$ | | | | | | | |
| Nitrate | 2.49 | 6.21 | 12.43 | 18.65 | 1.99 | 4.97 | 9.94 | 14.92 |
| Ammonium | 0.34 | 0.86 | 1.71 | 2.56 | 0.27 | 0.69 | 1.37 | 2.05 |
| Phosphorus | 1.26 | 1.26 | 1.26 | 1.26 | 1.58 | 1.58 | 1.58 | 1.58 |
| Potassium | 4.68 | 4.68 | 4.68 | 4.68 | 5.85 | 5.85 | 5.85 | 5.85 |
| Calcium | 3.55 | 3.55 | 3.55 | 3.55 | 3.55 | 3.55 | 3.55 | 3.55 |
| Magnesium | 1.56 | 1.56 | 1.56 | 1.56 | 1.56 | 1.56 | 1.56 | 1.56 |
| Sulfur | 1.63 | 1.63 | 1.63 | 1.63 | 2.04 | 2.04 | 2.04 | 2.04 |
| | $\mu mol \, L^{-1}$ | | | | | | | |
| Iron (Chelated) | 35.84 | 35.84 | 35.84 | 35.84 | 35.84 | 35.84 | 35.84 | 35.84 |
| Manganese | 7.29 | 7.29 | 7.29 | 7.29 | 7.29 | 7.29 | 7.29 | 7.29 |
| Boron | 27.78 | 27.78 | 27.78 | 27.78 | 27.78 | 27.78 | 27.78 | 27.78 |
| Zinc | 0.92 | 0.92 | 0.92 | 0.92 | 0.92 | 0.92 | 0.92 | 0.92 |
| Copper | 0.31 | 0.31 | 0.31 | 0.31 | 0.31 | 0.31 | 0.31 | 0.31 |
| Molybdenum | 0.63 | 0.63 | 0.63 | 0.63 | 0.63 | 0.63 | 0.63 | 0.63 |
| EC ($dS \, m^{-1}$) | 1.93 | 2.00 | 2.11 | 2.72 | 1.85 | 2.00 | 2.30 | 2.65 |

%: of standard N concentration; All nutrient solutions had their pH adjusted to 5.5. DAT: days after transplantation.

### 2.5. Data Collection

*Indexes of the 4th Leaf*. The 4th leaf of the potato plant, counting from the apex of the plant, is fully expanded and considered the standard leaf for morphometric measurements. Usually, this leaf has been used for the nutritional status analysis of the plant [35–39].

### 2.6. SPAD Readings of the 4th Leaf

The SPAD readings were measured with SPAD-502 chlorophyll meter (Konica Minolta) in the terminal leaflet of the 4th leaf. This non-destructive measurement was performed from 7:00 to 9:00 am.

*2.7. Nitrogen Balance Index, Chlorophyll, and Flavonoids of the 4th Leaf*

It was measured with the DUALEX Scientific in the terminal leaflet of the 4th leaf. This non-destructive measurement was performed from 7:00 to 9:00 am.

*2.8. Biometric Characteristics and Biomass of the 4th Leaf*

The leaf thickness was measured with an electronic digital micrometer, 0–25 mm range (Mitutoyo, Japan). We averaged three samples per leaf to obtain the final mean in each plot (6 plants). The leaf area was measured with the Delta-T Device WinDIAS 3 leaf image analysis system (https://www.dynamax.com/products/leaf-canopy-and-image-analysis/windias-3-image-analysis-system, accessed on 7 September 2022). Leaf length and width: these were obtained with a millimeter-graduated ruler. The leaflet numbers were counted manually on each plot. Leaf dry weight: after the evaluations, the fourth leaf was dried in a forced-air oven until it reached constant mass.

*2.9. Nitrate-N Petiole Sap-Test of the 4th Leaf*

This was measured with a LAQUA twin nitrate meter (Horiba) through the extraction of petiole sap.

*2.10. Minitubers Production*

Number and fresh weight of minitubers: The root system of the crop was monitored over time (with 7 days harvest intervals) at 33, 40, 47, 54, and 61 days after transplantation (DAT). After each harvest, the tubers were counted, and their mass was recorded.

*2.11. Biomass*

At 61 DAT, roots, stems, leaves, and total dry weight were evaluated. To evaluate the mass of the dry weight of the organs, plants were collected from the respective experimental units and packed in kraft paper bags. Subsequently, after separation, the organs (shoots and tubers) were placed in a forced-air oven at 70 °C until a constant mass was reached before the dry weight mass was determined for each organ. The mass of the total dry matter was obtained by the sum of the masses of the root, stems, leaves, and fourth leaf.

*2.12. Data Analysis*

Data were submitted for analysis of variance and regression. Means were compared by the Fisher's LSD test ($p < 0.05$) for the respective combinations of N concentrations and nutrient solutions. For the selection of indexes to diagnose N status and prognosticate the production of nuclear seed potatoes the Pearson's correlation ($p < 0.05$) analysis was performed. The best indexes were selected and proceeded regression analysis to establish the critical value. The models were chosen based on the biological logic, on the regression coefficient's significance, using the *t*-test ($p < 0.10$), and the coefficient of determination.

## 3. Results

*3.1. Indexes of the 4th Leaf*

There was a significant correlation for all indexes measured at 21 DAT with petiole sap nitrate in Otazú's nutrient solution (NS1). For modified Furlani's nutrient solution (NS2), there was a significant correlation between SPAD readings, flavanol index, area of the 4th leaf, leaflet numbers of the 4th leaf, and petiole sap nitrate (Table 3).

There was an effect of N concentrations for all selected indexes. To explain the relationship between a response variable (indexes) and a predictor variable (N), both linear and non-linear regression models were used. The lowest and highest adjusted coefficient of determination were 0.80 and 0.99, respectively. The maximum values for SPAD readings, area of the 4th leaf, 4th leaf length, 4th leaf dry weight, and 4th leaf thickness (LT) were: 39.47 (10.48 mmol$_c$ L$^{-1}$ of N), 196.87 cm$^2$ (13.81 mmol$_c$ L$^{-1}$ of N), 22.83 cm (12.18 mmol$_c$ L$^{-1}$ of N), 0.48 g (13.71 mmol$_c$ L$^{-1}$ of N), and 0.26 mm (13.75 mmol$_c$ L$^{-1}$ of N), respectively, for Otazú's nutrient solution. The maximum values for SPAD readings, 4th leaf area (4LA), 4th leaf

length (LL), and 4th leaf dry weight (4LDW) were: 40.59 (19.46 mmol$_c$ L$^{-1}$ of N), 188.61 cm$^2$ (12.30 mmol$_c$ L$^{-1}$ of N), 22.23 cm (14.77 mmol$_c$ L$^{-1}$ of N), and 0.46 g (13.89 mmol$_c$ L$^{-1}$ of N), respectively, for modified Furlani's nutrient solution (Table 4).

**Table 3.** Pearson's correlation coefficients between Nitrate-N Petiole Sap-Test and indexes measured on the 4th leaf of potato plants grown in aeroponics with Otazú's nutrient solution (NS1) and modified Furlani's nutrient solution (NS2).

| | Nitrate-N Petiole Sap-Test (mg L$^{-1}$) | |
|---|---|---|
| **Indexes at 21 DAT** | **NS1** | **NS2** |
| SPAD readings | 0.51 ** | 0.47 ** |
| Nitrogen Balance Index, NBI | 0.57 *** | 0.31 |
| Chlorophyll Index | 0.46 ** | −0.03 |
| Flavonol Index | −0.47 ** | −0.36 * |
| 4th Leaf Thickness (mm) | −0.45 ** | −0.29 |
| 4th Leaf Area (cm$^2$) | 0.75 *** | 0.48 ** |
| 4th Leaf length (cm) | 0.62 *** | 0.30 |
| 4th Leaf Width (cm) | 0.37 * | 0.13 |
| 4th Leaflet Numbers | 0.48 ** | 0.40 * |
| 4th Leaf Dry Weight (g) | 0.65 *** | 0.13 |

***, **, and *: Significant at the 0.001, 0.01, and 0.05 probability, respectively, by *t*-test. *n* = 36.

**Table 4.** Relationship between SPAD (S), leaf area (4LA), nitrogen balance index (NBI), leaf length (LL), leaf dry weight (LDW), and leaf thickness (LT) in the fourth leaf of potato at 21 DAT, and nitrogen concentrations (N) with the estimation of adjusted coefficients of determination ($\bar{r}^2$ *or* $\bar{R}^2$) for their nutrient solutions.

| Nutrient Solution (NS) | Equation | $\bar{r}^2$ or $\bar{R}^2$ | Maximum |
|---|---|---|---|
| | **SPAD** | | or Minimum Point |
| NS1 | $\hat{S} = 39.2245(1 - \exp(-0.5997N))$ | 0.95 | 10.48, 39.47 |
| NS2 | $\hat{S} = 24.2613 + 7.4044\sqrt{N} - 0.8393\,N$ | 0.99 | 19.46, 40.59 |
| | **4LA** (cm$^2$) | | |
| NS1 | $\hat{LA} = 91.3622 + 15.2807^{**}N - 0.5533^{**}N^2$ | 0.99 | 13.81, 196.87 |
| NS2 | $\hat{LA} = 42.4377 + 83.3509^{**}\sqrt{N} - 11.8819^{**}N$ | 0.99 | 12.30, 188.61 |
| | **NBI** | | |
| NS1 | $\hat{NBI} = 30.4422 + 0.6946^{**}N$ | 0.99 | − |
| NS2 | $\hat{NBI} = 42.8565 - (32.3961/N)$ | 0.83 | − |
| | **LL** (cm) | | |
| NS1 | $\hat{LL} = 6.4130 + 9.4074^{*}\sqrt{N} - 1.3477^{*}N$ | 0.99 | 12.18, 22.83 |
| NS2 | $\hat{LL} = 18.8892 + 0.4530^{**}N - 0.0153^{**}N^2$ | 0.99 | 14.77, 22.23 |
| | **4LDW** (g) | | |
| NS1 | $\hat{LDW} = 0.1999 + 4.0985^{*}N - 0.0015^{**}N^2$ | 0.98 | 13.71, 0.48 |
| NS2 | $\hat{LDW} = 0.4599 / \left[1 + \left(\frac{N-13.8851}{26.4515}\right)^2\right]$ | 0.89 | 13.89, 0.46 |
| | **LT** (mm) | | |
| NS1 | $\hat{LT} = 0.4032 - 0.0765^{**}\sqrt{N} + 0.0103^{**}N$ | 0.98 | 13.75, 0.26 |
| NS2 | $\hat{LT} = 0.2770 + (0.1745/N)$ | 0.80 | − |

** and *: Significant at 0.01 and 0.05 probability levels, respectively, by *t*-test. NS1: Otazú's Nutrient solution. NS2: Modified Furlani's Nutrient solution.

### 3.2. Prognosis Correlations

For plants fertilized with Otazú's nutrient solution, there were significant correlations between seed potato growth parameters (4th leaf thickness, 4th leaf area, leaf length, and 4th leaf dry weight) and NO$_3^-$ (measured at 21 DAT) and biomass (shoot, root or tuber number) accumulation at different harvests (Table 5). For plants fertilized with modified

Furlani's nutrient solution, there were significant correlations between seed potato growth parameters (4th leaf thickness, 4th leaf area, leaf length, and 4th leaf dry weight) and $NO_3^-$ (measured at 21 DAT) and biomass (shoot, root or tuber number) accumulation at different harvests (Table 6). These significant correlations can be also used to generate a prognosis for the final yield of potato minitubers. The fact that the same parameters were correlated with the leaf growth parameters (4th leaf thickness, 4th leaf area, leaf length, and 4th leaf dry weight) and the $NO_3^-$ concentration in both solutions suggest that well-fertilized plants up to 21 DAT will meet the requirements for mini tuber production as the $NO_3^-$ concentration itself is related to the leaf parameters that correlated with minituber yield at different harvests.

**Table 5.** Pearson's correlation coefficients between 4th leaf indexes at 21 DAT and potato plant parameters were evaluated at different harvests in response to Otazú's nutrient solution to generate a prognosis for the yield of nuclear seed potatoes.

|  | SPAD | NBI | CHL | FLV | LT | 4LA | LL | LW | LFN | 4LDW | $NO_3-$ |
|---|---|---|---|---|---|---|---|---|---|---|---|
| $\sum$FW | 0.15 | 0.08 | 0.08 | −0.15 | −0.28 | 0.49 ** | 0.44 ** | 0.17 | 0.28 | 0.49 ** | 0.38 * |
| $\sum$TN | 0.11 | 0.08 | 0.11 | −0.12 | −0.33 * | 0.38 * | 0.38 * | 0.10 | 0.14 | 0.37 * | 0.28 |
| RDW | 0.17 | 0.23 | 0.14 | −0.20 | −0.08 | 0.32 | 0.22 | 0.11 | 0.20 | 0.32 | 0.35 * |
| SDW | 0.39 * | 0.21 | 0.12 | −0.22 | −0.13 | 0.37 * | 0.41 * | 0.08 | 0.41 * | 0.43 ** | 0.51 ** |
| LDW | 0.36 * | 0.36 * | 0.25 | −0.26 | −0.26 | 0.48 ** | 0.45 ** | 0.17 | 0.37 * | 0.50 ** | 0.64 ** |
| TODW | 0.37 * | 0.33 * | 0.22 | −0.26 | −0.22 | 0.48 ** | 0.45 ** | 0.16 | 0.39 * | 0.50 ** | 0.62 ** |

** and *: Significant at the 0.01 and 0.05 probability levels, respectively, by *t*-test. *n* = 36. SPAD: SPAD readings, NBI: Nitrogen Balance Index, CHL: Chlorophyll index, FLV: Flavonol index, LT: Leaf thickness, 4LA: 4th leaf area, LL: Leaf length, LW: Leaf width, LFN: Leaflet number, 4LDW: 4th leaf dry weight, and $NO_3^-$: Nitrate-N Petiole Sap-Test, on 4th leaf of potato, $\sum$FW: Total fresh weight sum of all minituber harvests, $\sum$TN, Total sum of minituber numbers for all harvests, RDW: Root dry weight, SDW: Stem dry weight, LDW: Leaf dry weight, and TODW: Total dry weight.

**Table 6.** Pearson's correlation coefficients between 4th leaf indexes at 21 DAT and potato plant parameters evaluated at different harvests in response to modified Furlani's nutrient solution to generate a prognosis for the yield of nuclear seed potatoes.

|  | SPAD | NBI | CHL | FLV | LT | 4LA | LL | LW | LFN | 4LDW | $NO_3^-$ |
|---|---|---|---|---|---|---|---|---|---|---|---|
| $\sum$FW | −0.12 | −0.03 | −0.08 | −0.07 | 0.07 | 0.24 | 0.27 | −0.25 | 0.17 | 0.28 | 0.04 |
| $\sum$TN | −0.18 | −0.08 | −0.07 | 0.02 | 0.16 | 0.14 | 0.18 | −0.23 | 0.12 | 0.16 | −0.03 |
| RDW | 0.07 | −0.23 | 0.05 | 0.19 | −0.11 | 0.19 | 0.04 | −0.44 ** | 0.22 | 0.12 | 0.30 |
| SDW | 0.34 * | 0.11 | 0.15 | −0.15 | −0.17 | 0.37 * | 0.12 | −0.27 | 0.29 | 0.30 | 0.47 ** |
| LDW | 0.33 | −0.07 | 0.12 | −0.01 | −0.19 | 0.30 | 0.18 | −0.34 * | 0.27 | 0.18 | 0.39 * |
| TODW | 0.33 | −0.04 | 0.13 | −0.04 | −0.19 | 0.34 * | 0.17 | −0.35 * | 0.29 | 0.24 | 0.44 ** |

** and *: Significant at the 0.01 and 0.05 probability levels, respectively, by *t*-test. *n* = 36. SPAD: SPAD readings, NBI: Nitrogen Balance Index, CHL: Chlorophyll index, FLV: Flavonol index, LT: Leaf thickness, 4LA: 4th leaf area, LL: Leaf length, LW: Leaf width, LFN: Leaflet number, 4LDW: 4th leaf dry weight, and $NO_3^-$: Nitrate-N Petiole Sap-Test, on 4th leaf of potato, $\sum$FW: Total fresh weight sum of all minituber harvests, $\sum$TN, Total sum of minituber numbers for all harvests, RDW: Root dry weight, SDW: Stem dry weight, LDW: Leaf dry weight, and TODW: Total dry weight.

### 3.3. Minituber Numbers and Fresh Weight

At 33 DAT, there was no effect of N concentration on the minituber numbers (TNH33) for any of the nutrient solutions tested (Tables 5 and 6 and Figure 3A). However, there was an effect of N concentration on minituber numbers at 40, 47, 54, and 61 DAT, as well as total harvests for the nutrient solution proposed by Otazú [8] up to 100% of the crop-recommended dose, whereas the benefits of solution modified Furlani [3] appeared to wane after 50% of the recommended dose (Figure 3B–F). According to the total number of harvested tubers, Otazú's nutrient solution, dose 100%, and modified Furlani's nutrient solution, dose 50%, provided the best concentration of N for minitubers yield, while the modified Furlani's nutrient solution at 100% nitrogen concentration and 150% nitrogen

concentration of both nutrient solutions reduced minitubers yield. Potato plants are classified as moderately sensitive to salinity with a threshold for the saturated soil extract (ECe) of 1.7 dS m$^{-1}$ and irrigation water salinity (ECw) of 1.1 dS m$^{-1}$ [34]. Thus, based on the electrical conductivities of both Otazú's nutrient solution (ECw = 2.12 dS m$^{-1}$) and modified Furlani's nutrient solution (ECw = 2.72 dS m$^{-1}$) at 150% of the recommended dose, it is clear that both solutions had ECw above the salinity threshold (ECw = 1.1 dS m$^{-1}$) tolerated by potato plants. This threshold also explains why going over 50% of the recommended N dose on modified Furlani also exposed the plants to an ECw of 2.0 dS m$^{-1}$, also higher than the potato salinity threshold, bringing the minituber yield down at 40, 47, and 54 DAT (Figure 3B–D). At 61DAT, NS2 had half or less of the tuber numbers produced in previous harvests. The highest total number of tubers for Otazú's nutrient solution was at 100% while for modified Furlani's nutrient solution was at 50%, again due to the higher electrical conductivity of modified Furlani compared to Otazú's nutrient solution (Table 1).

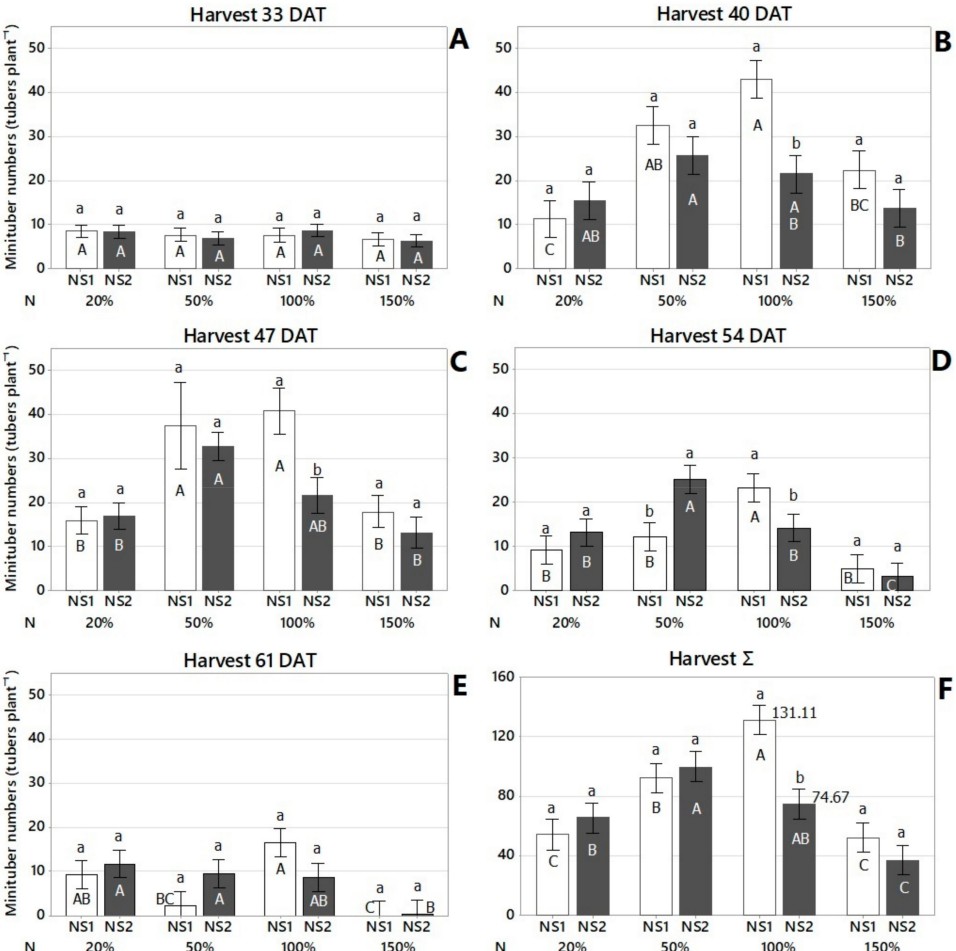

**Figure 3.** Effect of nutrient solution (NS) and nitrogen (N) concentration on minituber yield at 33 (**A**), 40 (**B**), 47 (**C**), 54 (**D**), 61 days after transplant—DAT (**E**), and total harvest (**F**). Lowercase letters indicate statistically different nutrient solutions (NS) within each nitrogen concentration (N) for each harvest time, according to Fisher's LSD test ($p < 0.05$). Uppercase letters indicate statistically different nitrogen concentrations (N) within each nutrient solution (NS), for each harvest time, according to Fisher's LSD test ($p < 0.05$). Interval bars represent standard errors ($n = 9$). NS1: Otazú's nutrient solution and NS2: modified Furlani's nutrient solution. N: nitrogen concentration of the crop-recommended dose to produce seed potatoes.

The minituber numbers at each harvest are provided in Figure 3A–E. However, based on the total sum of tubers provided for all harvests, Otazú's nutrient solution provided the highest

total number of tubers (131.11) at 100% N concentration, while modified Furlani's nutrient solution provided the highest total number of tubers (99.89) at 50% N concentration (Figure 3F).

The production of minitubers per plant increased for all treatments, reaching its high point at 40 and 47 DAT, and then decreased to the final harvest at 61 DAT for the nutrient solution proposed by Otazú [8], Figure 3B,C. For the modified-Furlani nutrient solution [3], the production of minitubers per plant increased from the beginning for all treatments, reaching its high point at 47 DAT, and then decreased until the final harvest at 61 DAT. This is another indication that the ECw of modified Furlani's nutrient solution provided more salt stress than the benefits provided by the N doses (Figure 3A–E).

Peak fresh weight minituber production was achieved at 40 and 47 DAT with 72.17 g plant$^{-1}$ and 44.65 g plant$^{-1}$, respectively, with the nutrient solution proposed by Otazú [8] at 100% of the proposed nitrogen concentration (Figure 4B,C). This concentration provided the maximum yields of minituber numbers with maximum average yields of 42.22 tubers plant$^{-1}$ at 40 DAT and 40.78 tubers plant$^{-1}$ at 47 DAT (Figure 3B,C). The 100% N (Otazú's nutrient solution) was significantly superior to the modified Furlani's nutrient solution for minituber fresh weight at 40 DAT, 61 DAT, and for total harvest (Figure 4B,E,F). There were no significant differences in the production of fresh weight minituber between treatments at 33 and 54 DAT. Except for the nitrogen concentration at 54 DAT (Figure 4A,D). 50% nitrogen concentration was significantly better than 150% nitrogen concentration in the nutrient solution modified from Furlani [3] for the production of fresh weight minituber. Within the nutrient solution proposed by Otazú [8], 100% nitrogen concentration outperformed the other nitrogen concentrations in terms of minituber fresh weight production (Figure 4D).

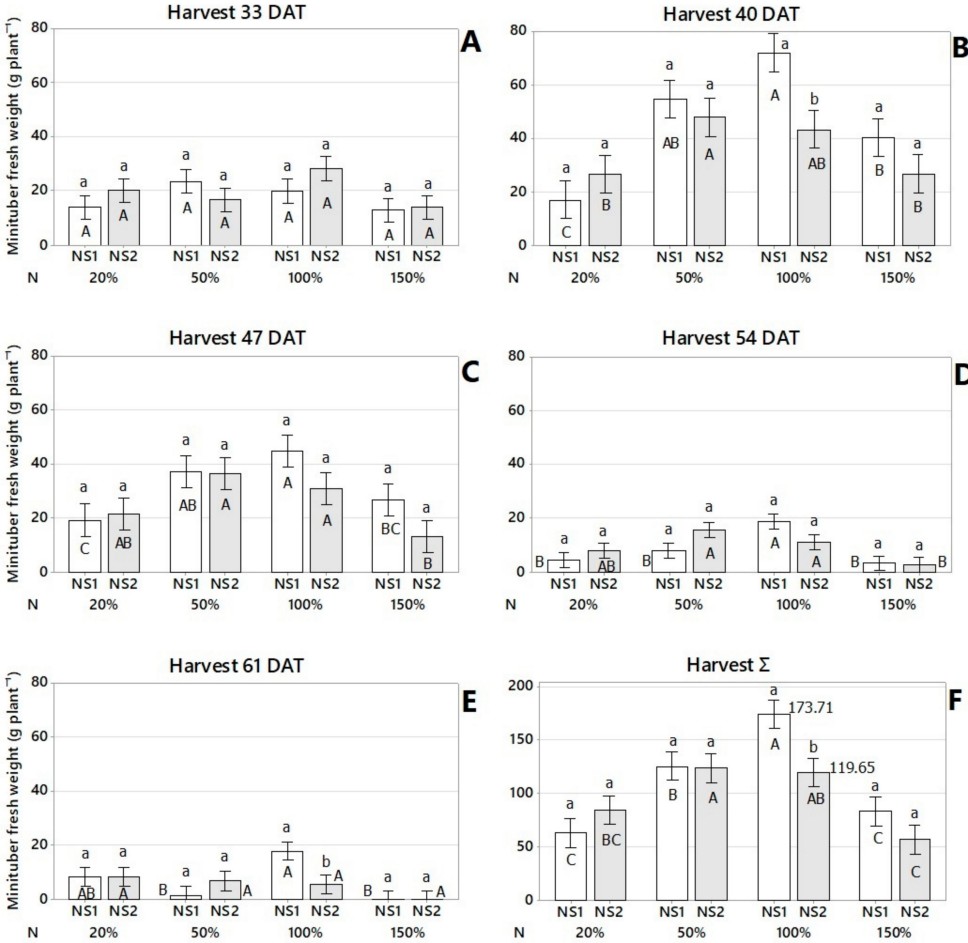

**Figure 4.** Effect of nutrient solution (NS) and nitrogen (N) concentration on minituber fresh weight at 33 (**A**), 40 (**B**), 47 (**C**), 54 (**D**), 61 days after transplant—DAT (**E**), and total harvest (**F**). Lowercase letters

indicate statistically different nutrient solutions (NS) within each nitrogen concentration (N) for each harvest time, according to Fisher's LSD test ($p < 0.05$). Uppercase letters indicate statistically different nitrogen concentrations (N) within each nutrient solution (NS), for each harvest time, according to Fisher's LSD test ($p < 0.05$). Interval bars represent standard errors ($n = 9$). NS1: nutrient solution proposed by Otazú (2010) and NS2: nutrient solution modified from Furlani (1998). N: nitrogen concentration of the crop-recommended dose to produce seed potatoes.

### 3.4. Minituber Numbers: Cross Diameters

Table 7 summarizes the effects of nutrient solution, nitrogen concentration, and cross diameter on average minituber number production within each harvest. 100% nitrogen concentration was significantly higher in minituber numbers yield than 20% and 150% nitrogen concentrations in the total sum of minitubers in all harvests with cross diameters of 16–23 mm, 10–16 mm, and 8–10 mm. When the nutrient solutions were compared within each cross diameter, harvest time, and nitrogen concentration, we discovered that the nutrient solution proposed by Otazú [8] produced significantly more minituber numbers than the nutrient solution modified from Furlani [3] 40 days after transplanting (DAT): cross diameters of 10–16 mm (nitrogen concentrations of 50%, 100%, and 150%), at 47 DAT cross diameters of 10–16 mm and 8–10 mm (100% nitrogen concentration), in the total of all harvests: cross diameters of 10–16 mm (nitrogen concentrations of 100% and 150%), 8–10 mm, and 6–8 mm (100% nitrogen concentration) (Table 7). The minimum and maximum transverse diameter size yields were 6 mm and 30 mm, respectively.

**Table 7.** Minituber numbers yield (tubers plant$^{-1}$) at 33, 40, 47, 54, 61 days after transplant (DAT), and total sum ($\sum$) of minitubers for the respective combinations of cross diameter (Ø), nutrient solution, and N concentration in an aeroponic system.

| Ø (mm) | NS1: Otazú (2010) | | | | | NS2: Modified Furlani (1998) | | | |
|---|---|---|---|---|---|---|---|---|---|
| | 20% | 50% | 100% | 150% | | 20% | 50% | 100% | 150% |
| | | | | 33 DAT | | | | | |
| 23–30 | 1.11bB | 2.44aAB | 1.22bcB | 3.00aA | | 1.78abA | 2.67aA | 1.56bcA | 2.00aA |
| 16–23 | 1.56bB | 3.33aA | 3.11aAB | 2.22abAB | | 3.22aAB | 2.78aAB | 4.33aA | 2.00aB |
| 10–16 | 4.22aA | 1.89aB | 2.78abAB | 1.22bcB | | 2.89aA | 1.22abA | 2.78abA | 1.33aA |
| 8–10 | 0.89bA | 0.00bA | 0.44cA | 0.11cA | | 0.33bA | 0.11bA | 0.00cA | 0.33aA |
| 6–8 | 0.67bA | 0.00bA | 0.00cA | 0.11cA | | 0.11bA | 0.11bA | 0.00cA | 0.67aA |
| | | | | 40 DAT | | | | | |
| 23–30 | 0.00bA | 1.22cA | 0.89cA | 1.56bA | | 0.44bA | 2.56bcA | 2.33bcA | 0.44bA |
| 16–23 | 1.44bB | 6.78bA | 7.56bA | 3.67bAB | | 2.22bB | 6.22bA | 5.11bAB | 3.56abAB |
| 10–16 | 6.56aD | 18.78aB * | 26.11aA * | 13.00aC* | | 9.33aBC | 13.56aA | 11.22aAB | 6.44aC |
| 8–10 | 2.33bB | 4.22bcAB | 7.00bA * | 3.22bAB | | 2.67bA | 3.33bcA | 2.89bcA | 2.67abA |
| 6–8 | 1.00bA | 1.56cA | 1.67cA | 1.00bA | | 0.78bA | 0.11cA | 0.00cA | 0.67bA |
| | | | | 47 DAT | | | | | |
| 23–30 | 0.00bA | 0.00dA | 0.67cA | 0.11bA | | 0.00bA | 0.00cA | 0.00bA | 0.00bA |
| 16–23 | 1.00bA | 1.56cdA | 1.67cA | 2.22bA | | 1.22bA | 1.78cA | 1.89bA | 0.44bA |
| 10–16 | 9.78aB | 19.11aA | 21.89aA * | 9.67aB | | 10.22aBC | 16.56aA | 13.56aAB | 5.89aC |
| 8–10 | 3.11bB | 12.11bA | 13.00bA * | 4.56bB | | 4.11bB | 9.67bA | 4.44bB | 3.89abB |
| 6–8 | 2.00bA | 4.67cA | 3.56cA | 1.33bA | | 1.33bA | 4.56cA | 1.67bA | 2.89abA |
| | | | | 54 DAT | | | | | |
| 23–30 | 0.11bA | 0.00bA | 0.00bA | 0.00aA | | 0.00bA | 0.00cA | 0.00bA | 0.00aA |
| 16–23 | 0.00bA | 0.22bA | 1.44bA | 0.11aA | | 0.11bA | 0.11cA | 0.00bA | 0.00aA |
| 10–16 | 1.56bB | 3.56aB | 7.00aA | 1.22aB | | 3.56aBC | 6.78bA * | 5.89aAB | 1.44aC |
| 8–10 | 2.78abB | 4.44aAB | 6.33aA | 1.78aB | | 4.00aB | 10.67aA * | 4.78aB | 0.33aC |
| 6–8 | 4.67aB | 3.89aB | 8.33aA * | 1.89aB | | 5.33aAB | 7.67bA * | 3.44aBC | 1.22aC |

**Table 7.** *Cont.*

| Ø (mm) | NS1: Otazú (2010) | | | | | NS2: Modified Furlani (1998) | | | |
|---|---|---|---|---|---|---|---|---|---|
| | 20% | 50% | 100% | 150% | | 20% | 50% | 100% | 150% |
| | | | | | 61 DAT | | | | |
| 23–30 | 0.00aA | 0.00aA | 0.22aA | 0.00aA | | 0.11aA | 0.00aA | 0.22aA | 0.00aA |
| 16–23 | 0.44aA | 0.11aA | 0.67aA | 0.00aA | | 0.22aA | 0.22aA | 0.00aA | 0.00aA |
| 10–16 | 3.56aA | 0.33aA | 7.89aA | 0.00aA | | 3.44aA | 2.56aA | 2.67aA | 0.00aA |
| 8–10 | 2.89aA | 0.38aA | 3.67aA | 0.00aA | | 3.44aA | 3.11aA | 2.22aA | 0.00aA |
| 6–8 | 2.44aA | 1.44aA | 4.00aA | 0.00aA | | 4.44aA | 3.56aA | 3.67aA | 0.33aA |
| | | | | | Harvest $\sum$ | | | | |
| 23–30 | 1.22cA | 3.67cA | 3.00dA | 4.67bA | | 2.33cA | 5.22dA | 4.11cA | 2.44bA |
| 16–23 | 4.44bcB | 12.00cAB | 14.44cA | 8.22bAB | | 7.00bcA | 11.11cdA | 11.33bcA | 6.00bA |
| 10–16 | 25.67aC | 43.67aB | 65.67aA * | 25.11aC * | | 29.44aB | 40.67aA | 36.11aAB | 15.11aC |
| 8–10 | 12.00bC | 21.11bB | 30.44bA * | 9.67bC | | 14.56bB | 26.89bA | 14.33bB | 7.22abB |
| 6–8 | 10.78bAB | 11.56cAB | 17.56cA * | 4.33bB | | 12.00bAB | 16.00cA | 8.78bcAB | 5.78bB |

Means followed by the same letter are not significantly different, lower-case letters indicate the effect of cross diameter and upper-case letters indicate the effect of N concentration at each harvest date within each nutrient solution by Fisher LSD test ($p < 0.05$). *: means followed by the star are significantly different, indicating the effect of nutrient solution at each N concentration and cross diameter within each harvest time by Fisher LSD test ($p < 0.05$).

At all nitrogen concentrations, approximately 48% of minituber yields had a cross diameter of 10–16 mm for the nutrient solution proposed by Otazú [8]. On the other hand, minituber yield with a cross diameter of 10–16 mm for nutrient solution modified from Furlani [3], averaged 44% for all nitrogen concentrations.

## 4. Discussion

The early prediction of nitrogen indices in the plant for seed potato production is critical for perfect nitrogen management, use, and efficiency. The researcher or extensionist will be able to adjust nitrogen fertilization by using index tools such as the Nitrogen Balance Index (NBI), Nitrate-N Petiole Sap-Test, and SPAD meter to predict seed potato production (amount and time). The laboratory method of determining the plant's nitrogen status usually takes a week to complete and, for aeroponic potatoes, it may be too long. [40]. Gerendás and Pieper [40] studied different non-destructive indexes to determine nitrogen status for seed potato production in three cultivars of nursery potatoes: Arnika, Baltica, and Secura. The Nitrate-N Petiole Sap-Test, according to the authors, can be used to monitor the nitrogen status of nursery potatoes.

Opto-electronic tests, used to measure the petiole sap nitrate ($NO_3^-$), can aid in rapid decision-making under both field and greenhouse conditions. The diagnosis of N status in the plants can be measured indirectly and non-destructively through SPAD readings and Nitrate-N Petiole Sap-Test [16,41,42].

Although the diagnosis of the N status is important for potato cultivation because nitrogen (N) plays an important role in the differentiation and growth of tubers, excess N may delay differentiation and initial growth of tubers, stimulating shoot growth [43,44]. If a high N supply is provided before tuberization it may result in stolon elongation and delay the onset of tuberization. This effect may be due to changes in the levels of growth regulators such as gibberellic acid and abscisic acid [45–47].

The N concentrations of 14.20 mmol$_c$ L$^{-1}$ up to 21 DAT and 9.80 mmol$_c$ L$^{-1}$ N from 22 to 61 DAT (100% of N) in Otazú's nutrient solution were similar to the concentrations of N in modified Furlani's nutrient solution (14 mmol$_c$ L$^{-1}$ up to 21 DAT and 11 mmol$_c$ L$^{-1}$ from 22 to 61 DAT). Thus, the N concentration of both solutions does not justify the significant difference in total tuber numbers. Instead, the results are justified by the fact that modified Furlani's nutrient solution uses salts with a higher salt index and had a higher electrical conductivity (ECw) than Otazú's nutrient solution at all concentrations tested (Table 1). When that ECw surpassed the salinity threshold of potato plants (ECw = 1.1) the tuber numbers started to decrease. Our ECw is a bit higher than the threshold ECw cited

previously [34] but different responses, and thresholds, to salinity are expected for different cultivars of the same species, as reported for five commercial strawberry cultivars tested at different salinities of irrigation water [48].

Up to 21 days after transplanting (DAT), the ammonium:nitrate ratio for the nutrient solution proposed by Otazú [8] was 0.45 (45:55, 45% ammonium and 55% nitrate) at 20% nitrogen concentration and 0.31 (31:69, 31% ammonium and 69% nitrate) at 50%, 100%, and 150% nitrogen concentration. The ammonium:nitrate ratio was 0.22 for all nitrogen concentrations from 22 to 61 DAT (Table 1). The ammonium:nitrate ratio to nutrient solution modified Furlani [3] was 0.12 (12:88) at all nitrogen concentrations and throughout the growth cycle, plant development, and minituber production (Table 1). There were no studies that evaluated the effect of the ammonium:nitrate ratio on the productivity of minituber seed potatoes in an aeroponic system. The higher ammonium:nitrate ratio of 0.22 produced more seed-potato minitubers in our study than the lower ammonium:nitrate ratio of 0.12. This demonstrates that, despite having a higher ammonium:nitrate ratio, Otazú's nutrient solution had lower electrical conductivity values that favored minitubers production. When the nutrient solution modified from Furlani [3] was used, higher values of electrical conductivity and higher salt indexes hampered the production of minituber seed potatoes. Serio et al. [49], on the other hand, used the trough bench sub-irrigation system to evaluate potato production in pots over two seasons with ammonium:nitrate percentage ratios of 100:0, 50:50, and 0:100. The authors concluded that the ammonium:nitrate percentage ratios of 50:50 or 0:100 increased potato yield. Cao and Tibbitts [50] confirmed the effect of ammonium:nitrate percentages of 0:100, 20:80, 40:60, 60:40, 80:20, and 100:0 in the first experiment. In the second experiment, the authors looked at ammonium:nitrate ratios of 0:100, 4:96, 8:92, 12:88, 16:84, and 20:80. The total nitrogen used in both experiments was 4 mmol $L^{-1}$. The authors used a non-recirculating nutrient film system (NFT) and harvested the potato plants at 35 DAT. They concluded that small concentrations of ammonium in total N could help boost nitrogen uptake and potato yield.

The total production of minitubers per plant with nutrient solution proposed by Otazú [8] was almost twice as high as that achieved with nutrient solution modified Furlani [3] (131.11 vs. 74.67) at 100% of the recommended dose. The reduction in tuber numbers is another sign that the higher electrical conductivity of modified Furlani's nutrient solution (ECw = 2.3 dS $m^{-1}$) compared to Otazú's nutrient solution (ECw = 1.35 dS $m^{-1}$), from 22–61 DAT and at 100% of the recommended dose, stressed the potato plants causing a significant reduction in tuber numbers (Figure 3F). Excess salinity has been reported to cause a significant reduction in fruit number in both strawberry and melon crops [48,51] and reduced the number of tubers in Jerusalem artichoke in approximately 50% when the salinity of the irrigation water increased from 1.2 to 12 dS $m^{-1}$ [52,53].

## 5. Conclusions

The 4th leaf area, 4th leaflet number, flavonol index, and SPAD index were useful N diagnosing parameters in potato plants for both nutrient solutions. The following indexes can be used to prognosticate minituber potato yield: leaf thickness, leaf area, leaf length, and leaf dry weight (Otazú's nutrient solution). Shoot dry weight, leaf dry weight, and total dry biomass can be used to prognosticate seed potato minituber production in both nutrient solutions. Because Otazú's nutrient solution had an $EC_w$ 31% lower than the modified Furlani's nutrient solution (1.6 vs. 2.1 dS $m^{-1}$) at 100% of the standard N concentration, and potatoes have a threshold salinity of ECw = 1.1 dS $m^{-1}$, the average fresh weight and the total number of minitubers achieved with Otazú's solution was 45% and 76% higher, respectively, than those obtained with the modified Furlani's solution. This indicates that a nutrient solution with salinity closer to the salinity threshold of potato should be preferred for the highest yield of minitubers. An ECw higher than the potato salinity threshold also explains why both nutrient solutions had significant drops in tuber yields at 150% of the recommended N dose.

The higher ammonium:nitrate ratio did not affect the productivity of minituber seed potatoes but the high ECw, and the high salt index of the salt components used to make the modified Furlani's nutrient solution may have hampered minituber seed potato production. Electrical conductivity of 1.6 dS m$^{-1}$ up to 21 DAT and 1.35 dS m$^{-1}$ from 22 DAT to 61 DAT produced significantly higher numbers and fresh weight of minitubers when Otazú's nutrient solution was used at 100% N concentration.

Based on the findings of this study, Otazú's nutrient solution at 100% of the standard N concentration was the best for producing nuclear potato minitubers in the International Potato Center's aeroponic system. Future studies should evaluate nutrient solutions that provide all major macro and micronutrients but that use salts with a low salt index to stay within the limits of the salinity threshold of potatoes or other crops produced in aeroponic systems. Additionally, nutrient solutions with lower salt indexes and solution electrical conductivities, and involving multiple seed-potato cultivars need to be investigated further.

**Author Contributions:** Conceptualization, J.B.S.F. and P.C.R.F.; methodology, J.B.S.F. and P.C.R.F.; formal analysis, J.B.S.F. and P.R.C.; investigation, J.B.S.F.; resources, P.C.R.F.; data curation, J.B.S.F.; writing—original draft preparation, J.B.S.F., J.F.S.F. and E.C.; writing—review and editing, J.B.S.F. and J.F.S.F.; visualization, J.B.S.F. and J.F.S.F.; supervision, J.B.S.F.; project administration, J.B.S.F. and P.C.R.F.; funding acquisition, J.B.S.F. and P.C.R.F. All authors have read and agreed to the published version of the manuscript.

**Funding:** This research was funded by the Brazilian National Council for Scientific and Technological Development (CNPq), process 303448/2018-0, the Foundation of Support Research of the State of Minas Gerais, Brazil (FAPEMIG), and the Coordenação de Aperfeiçoamento de Pessoal de Nível Superior, Brazil (CAPES), Finance Code 001.

**Data Availability Statement:** The new data created here was for the sole purpose of testing our treatments and hypotheses and not to be entered in any public database or archives. Thus, Data sharing does not apply to this article.

**Acknowledgments:** We thank the Brazilian National Council for Scientific and Technological Development (CNPq), process 303448/2018-0, and the Foundation of Support Research of the State of Minas Gerais, Brazil (FAPEMIG) for their financial support. Additionally, this study was financed in part by the Coordenação de Aperfeiçoamento de Pessoal de Nível Superior, Brazil (CAPES), Finance Code 001.

**Conflicts of Interest:** The authors declare no conflict of interest. The use of trade, firm, or corporation names in this publication is simply for the convenience of the reader and does not constitute an official endorsement by the authors, the Federal University of Viçosa, the University of California Riverside, or the United States Department of Agriculture (USDA) over any similar product or service and does not exclude other products or services that may be suitable. The USDA is an equal-opportunity employer.

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
