# Peer review of "Optimal Nutrient Solution and Dose for the Yield of Nuclear Seed Potatoes under Aeroponics"

_agronomy, doi:10.3390/agronomy12112820_

Round 1

Reviewer 1 Report

This is a very well written paper. However, from the perspective of aeroponics, there are still the following problems:
1. In fact, different spray intervals and spray times have a great influence on potato growth. The author would do well to state the reasons for the choice of these parameters.
2. Droplet size is the key parameter of the fog cultivation system. The effect of aeroponics varies greatly under different droplet sizes, and it is best to add the droplet size information to the manuscript.
3. All conclusions of the paper are drawn under specific spray parameters and droplet sizes, which must be made clear in the conclusions.

Author Response

Dear Editor:

We want to thank you and the reviewers for taking the time to read and comment on our manuscript. The point-by-point responses for each reviewer are listed below in green.

Reviewer 1

This is a very well written paper. However, from the perspective of aeroponics, there are still the following problems:
1. In fact, different spray intervals and spray times have a great influence on potato growth. The authors would do well to state the reasons for the choice of these parameters.

This information was added on page 4 lines 121-138 in the newly revised version.

  1. Droplet size is the key parameter of the fog cultivation system. The effect of aeroponics varies greatly under different droplet sizes, and it is best to add the droplet size information to the manuscript.

This information was also added on page 4 lines 121-138 in the newly revised version.

  1. All conclusions of the paper are drawn under specific spray parameters and droplet sizes, which must be made clear in the conclusions.

We would like to remind the reviewer that the spray parameters and droplet sizes were established in a previous publication and were not the focus of this research. To be clear, our conclusions addressed the following points:

  1. The best parameters to prognosticate minitubers potato yield.
  2. The fact that the salinities of the solutions were different and the modified Furlani’s nutrient solution had salinity or electrical conductivity (EC) of 2.1 dS/m (31% higher than Otazu’s solution) thus causing a significant reduction in minitubers numbers and fresh weight. In the same paragraph, we also concluded that a nutrient solution with an EC close to that of potato salinity tolerance threshold would be more appropriate for seed-potato minitubers production under aeroponics. We modified the text to make these points clearer from lines 585 to 589.
  3. We also concluded that, due to the high ECw’s of both nutrient solutions at 150% of the recommended N dose recommended for potato, both resulted in significant reductions in minitubers yield.
  4. We concluded also that the higher ammonium:nitrate ration had no effect on minitubers yield but the ECw and the high salt index of the salt components of the modified Furlani’s solution were probably responsible for the reduction in minitubers yield and fresh weight when compared to Otazu’s solution.
  5. We also removed the part on the ammonium:nitrate ratio from the conclusion after concluding that this ratio had no effect on the results.
  6. We concluded that Otazu’s solution was better than the modified Furlani’s solution to produce nuclear potato minitubers in the International Potato Center’s aeroponic system.

Reviewer 2 Report

The work concerns the assesement  of two nutrient and possibility of forecasting the degree of nitrogen nutrition of potato plants grown in aeroponic. 

The manusctript is very well written and requies no major correction

My doubt is : why such high concentration (150%) were used, knowing from the beginning that salinity would be too high for potato plants, especially in case of Furlani.

Detailed comments

1. Lines 45-46 - incomprehensible text

2. Lines 85-88 (Materials and Methods) maximum and minimum temperature and relative humidity during the experiment- too large spread, it should be detailed.

3. Lines 193-194. The tuber diameter data should be rather  in Result section

Author Response

Dear Editor:

We want to thank you and the reviewers for taking the time to read and comment on our manuscript. The point-by-point responses for each reviewer are listed below in green.

Reviewer 2

The work concerns the assessment of two nutrient and possibility of forecasting the degree of nitrogen nutrition of potato plants grown in aeroponic. 

The manuscript is very well written and requires no major correction

My doubt is: why such high concentration (150%) was used, knowing from the beginning that salinity would be too high for potato plants, especially in case of Furlani.

When looking for optimal nitrogen dose values, we must consider extreme values, both up and down, despite the hypothesis that values above and below those established by other authors can harm seed potato productivity. Also, by using higher concentrations than recommended we can establish that the recommended dose still applies for the new cultivars used as these recommended doses were established several years ago but they did not consider salt indexes of salts or the electrical conductivity (EC) of the nutrient solution based on the potato threshold for salinity. Furthermore, when we use linear regression to analyze these data statistically, we have the assurance that the highest dose decreases productivity (due to salinity) and thus recommend producers not to surpass that dose, even with recent commercial cultivars (more fertilizer leads to salinity and not to higher yields in this case). I hope we made it clearer why using a dose higher than 100% of recommended.

Detailed comments

  1. Lines 45-46 - incomprehensible text

We could not quite identify the source of confusion in the text range referred by the reviewer.  However, we changed the text in the hope of achieving better clarity, as follows (Lines 44-51 of the newly revised version contains the whole modified sentence): “A five-year study on irrigated potato production system in a sandy loam soil evaluated nitrogen (N) sources such as ammonium sulfate, ammonium nitrate, urea, and calcium nitrate to maximize potato yield [9]. At the end, Bundy and co-workers [9] concluded that ammonium nitrate was the best N source for potato production in an irrigated system. Silva et al. [10] evaluated calcium nitrate and urea as N sources in a hydroponic system. Contrary to Bundy et al. [9], they concluded that the nitric, rather than the ammoniacal, source was more suitable for potato production in hydroponic systems.”

  1. Lines 85-88 (Materials and Methods) maximum and minimum temperature and relative humidity during the experiment- too large spread, it should be detailed.

This was added to page 2-3, lines 88-96 in the newly revised version.

  1. Lines 193-194. The tuber diameter data should be rather in Result section

This has been moved to page 14, lines 398-399 in the newly revised version.

Please let me know if you still have any questions or concerns.

Thank you for considering the submission of this manuscript.

Respectfully,

Jaime B. Silva Filho and Jorge F.S. Ferreira

Plant Physiologists, Ph.D.

Corresponding authors